# UNDERSTANDING REASONING IN CHAIN-OF-THOUGHT FROM THE HOPFIELDIAN VIEW

## ABSTRACT

Large Language Models have demonstrated remarkable abilities across various tasks, with Chain-of-Thought (CoT) prompting emerging as a key technique to enhance reasoning capabilities. However, existing research primarily focuses on improving performance, lacking a comprehensive framework to explain and understand the fundamental factors behind CoT's success. To bridge this gap, we introduce a novel perspective grounded in the Hopfieldian view of cognition in cognitive neuroscience. We establish a connection between CoT reasoning and key cognitive elements such as stimuli, actions, neural populations, and representation spaces. From our view, we can understand the reasoning process as the movement between these representation spaces. Building on this insight, we develop a method for localizing reasoning errors in the response of CoTs. Moreover, we propose the Representation-of-Thought (RoT) framework, which leverages the robustness of low-dimensional representation spaces to enhance the robustness of the reasoning process in CoTs. Experimental results demonstrate that RoT improves the robustness and interpretability of CoT reasoning while offering fine-grained control over the reasoning process.

## 1 INTRODUCTION

Large Language Models (LLMs) have demonstrated exceptional capabilities in following the natural language instructions (Ouyang et al., 2022; Jin et al., 2024) and excelling across a variety of downstream tasks (Hu et al., 2023a; Zhang et al., 2023). As reasoning skills are crucial for tasks such as commonsense and mathematical reasoning (Rae et al., 2021), there is a growing focus on enhancing these capabilities. One prominent approach is Chain-of-Thought (CoT) prompting (Wei et al., 2022; Kojima et al., 2022), a simple yet highly effective technique to unleash the reasoning capability of LLMs. However, despite its success, a natural and fundamental research question remains: *How does the reasoning capability emerge through CoT prompting?*

Numerous studies have sought to identify the key factors or elements that enable CoT to enhance the reasoning capabilities of LLMs (Kojima et al., 2022; Wang et al., 2023a; Tang et al., 2023; Merrill & Sabharwal, 2023). Some works focus on improving CoT reasoning through query-based corrections (Kim et al., 2023), knowledge-enhanced frameworks (Zhao et al., 2023), and symbolic reasoning chains for faithful CoT (Lyu et al., 2023; Lanham et al., 2023). Other research has examined how the sequence of demonstrations, random labels (Min et al., 2022), or even meaningless tokens (Pfau et al., 2024) can positively influence reasoning performance. However, these works primarily focus on improving the model's reasoning performance, and they do not provide a comprehensive framework to explain the underlying factors driving CoT's success.

To understand the reasoning process in CoTs more deeply, we draw inspiration from cognitive neuroscience, specifically the relationship between cognition and brain function. In this field, the Hopfieldian view (Hopfield, 1982) and the Sherringtonian view (Sherrington, 1906) represent two different ways of understanding neural computational models and cognitive mechanisms. While the Sherringtonian view of cognitive explanation focuses on specific connections between neurons in the brain, the Hopfieldian view emphasizes distributed computation across neural populations, where information is not encoded by a single neuron but rather by the cooperative activity of many neurons. This perspective is particularly suited to explaining complex cognitive functions like memory storage, pattern recognition, and reasoning. Thus, the Hopfieldian view is generally considered more advanced than the Sherringtonian view, especially in the context of explaining distributed computation

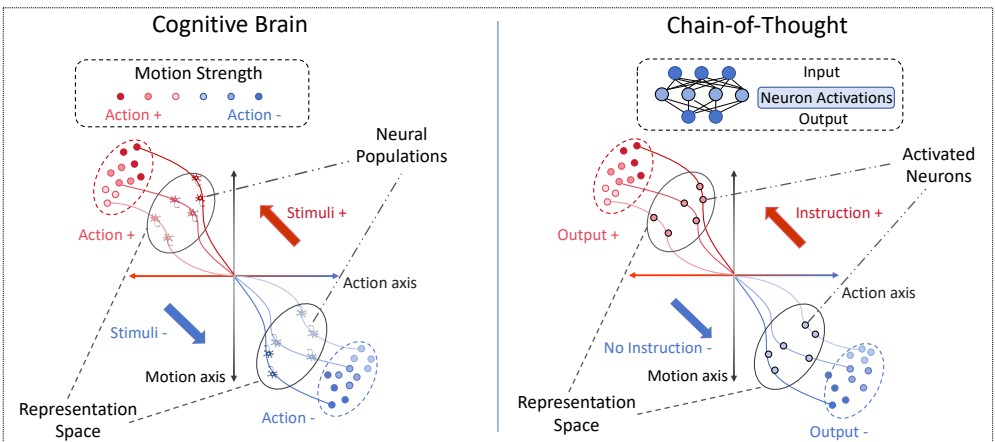

Figure 1: Illustration of the emergence of cognition in the brain and CoT reasoning from the Hopfieldian view.

and the dynamics of neural networks (Barack & Krakauer, 2021). Based on these, a natural question is: *whether we can understand the reasoning in CoTs from the Hopfieldian view of cognition?*

The Hopfieldian view explains the production of behavioral actions as emerging from transformations or movements within neural populations in response to *stimuli* in the brain (Barack & Krakauer, 2021) (cf. Figure 1). This perspective approaches cognition *at the level of representations*, disregarding the detailed roles of individual molecules, cells, and circuits, thus allowing the potential for a more conceptual and semantic understanding of complex cognitive systems. Viewing the CoT-triggered reasoning process in LLMs through this lens is intuitive: CoT prompting induces shifts in the model's trajectory in much the same way that external stimuli shape cognitive responses, driving representation changes without altering the underlying system. Specifically, similar to the Hopfieldian mechanism, where the shift or movement in neural populations happens during cognition itself, CoT influences reasoning during inference, controlling the logical steps without modifying the model's parameters.

Given the parallels between the CoT-triggered reasoning process and the Hopfieldian view of cognition in the brain, we first establish a connection between these two by aligning key elements: *stimuli and actions*, *neural populations*, and *representation spaces*. Particularly, we provide a general framework for identifying the "representation spaces" of the "stimuli" given by CoTs. We conceptualize the reasoning process elicited by CoT prompting as movement between representation spaces, enabling us to improve and deepen our understanding of CoTs. Based on these connections, we then leverage the strength of the Hopfieldian view to improve or further understand CoTs. Specifically, by leveraging the "representation spaces" in CoTs, we develop a method for localizing the reasoning error in the responses. Moreover, by leveraging the robustness of low-dimensional representation spaces, we propose a new framework, namely Representation-of-Thought (RoT), which enhances the robustness of CoTs. We summarize the key contributions of our work as follows:

1. We establish a connection between the reasoning process in CoTs and the Hopfieldian view of cognition, grounded in cognitive neuroscience, to identify the key factors driving CoT's success in zero-shot and few-shot settings. To the best of our knowledge, this is the first known attempt to leverage cognitive science for CoT interpretability by associating its core elements with the Hopfieldian framework.

2. Based on these connections, we leverage the strength of the Hopfieldian view to understand and further improve CoTs. We first consider how to localize the reasoning error based on the low-dimensional representation spaces. Then, by leveraging the robustness of the Hopfieldian view, we propose a new framework, RoT, to enhance the robustness of CoTs' performance.

3. Comprehensive experiments on three tasks, including arithmetic reasoning, commonsense reasoning, and symbolic reasoning, reveal that our framework can provide intuitive and interpretable analysis, allowing error tracing and control for CoT reasoning.

## 2   RELATED WORK

**Chain-of-Thought (CoT).**   The CoT is a prompting technique that engages LLMs in step-by-step reasoning rather than directly providing the answers (Nye et al., 2021). Studies have shown that introducing intermediate steps or learning from demonstrations can significantly improve the reasoning performance of LLMs (Wei et al., 2022; Kojima et al., 2022). Given the success of CoT, numerous studies have explored its application to a variety of complex problems, including arithmetic, commonsense, symbolic reasoning (Wang et al., 2023c; Zhou et al., 2023; Wang & Zhou, 2024), and logic tasks (Creswell & Shanahan, 2022; Pan et al., 2023; Weng et al., 2023). Recently, numerous endeavors have been made to enhance the reasoning capabilities in LLMs (Wang et al., 2023a; Dutta et al., 2024). For example, Kim et al. (2023) proposed a query-based approach to correct erroneous reasoning steps within a CoT. Zhao et al. (2023) introduced a knowledge-enhanced method to improve the factual correctness for multi-pole open-domain QA tasks. Lyu et al. (2023) developed "faithful CoT", *i.e.,* a framework that first translates natural language queries into symbolic reasoning chains and then solves the problem using CoT. Additionally, several studies have also focused on the sequence and quantity of demonstrations within the context, investigating their contributions to the final reasoning performance. For this, Min et al. (2022) discovered that even random labels or ineffective reasoning steps can still improve the model's reasoning performance. Lanham et al. (2023) demonstrated the impact of intervening in the CoT process by adding mistakes or paraphrases. Pfau et al. (2024) showed that using meaningless filler tokens in place of a chain-of-thought can surprisingly boost reasoning performance. However, these studies primarily focused on how to improve the CoT's reasoning performance and do not provide a framework to analyze the fundamental reasons, *i.e.,* how does the reasoning capability emerge through CoT? Dutta et al. (2024) investigates the neural sub-structures within LLMs that manifest Chain-of-Thought (CoT) reasoning on the Llama-2-7B model. Similarly, Rai & Yao (2024) explores neurons in the feed-forward layers of LLMs to analyze their arithmetic reasoning capabilities on the Llama-2-7B model. Both studies are grounded in the Sherringtonian view of neural activity. In contrast, we adopt the Hopfieldian perspective to bridge this gap, focusing on representations rather than individual neurons. We apply our approach across three different downstream tasks and can further extend our analysis to larger models like Llama-2-70B.

**Interpretability of LLMs.**   Interpretability plays a key role in a deeper understanding of LLMs to identify potential risks and better meet human requirements (Zou et al., 2023). Common interpretability strategies include (i) Salience maps, which rely on highlighting the regions in the input that are attended by the model (Simonyan et al., 2014; Smilkov et al., 2017; Clark et al., 2019; Hu et al., 2023c;b; Lai et al., 2024); (ii) Feature visualization, which creates representative inputs indicative of particular neurons' activations (Szegedy et al., 2014; Nguyen et al., 2016; Fong & Vedaldi, 2018; Nguyen et al., 2019); and (iii) Mechanistic interpretability, which employs reverse-engineering tools to explain networks based on circuits and node-to-node connections (Olah et al., 2020; Olsson et al., 2022; Wang et al., 2023b). However, these methods often require substantial human intervention and are limited in terms of scalability or interpretability, especially for the large language models (Fong & Vedaldi, 2018; Jain & Wallace, 2019; Hu et al., 2024). Thus, these methods cannot be directly used to interpret CoT reasoning. Additionally, most current approaches focus on representation-level analysis without considering how these representations connect to concepts learned during pre-training (Bricken et al., 2023; Templeton et al., 2024). Other works investigate the localization and representation of concepts in the network (Kim et al., 2018; Li et al., 2024), linear classifier probing to uncover input properties (Belinkov, 2022), fact localization and editing (Meng et al., 2022; Zhong et al., 2023; Cheng et al., 2024a;b), concept erasure (Shao et al., 2023; Gandikota et al., 2023), and corrective analysis (Burns et al., 2023), *etc.* These observations are aligned with RepE (Zou et al., 2023), which emphasized the nearly linear nature of LLM representations (Park et al., 2024). However, none of these approaches directly address the inner workings of CoT reasoning. While recent work has begun exploring connections between LLM interpretability and cognitive neuroscience (Vilas et al., 2024). However, it does not discuss the Hopfieldian view and also does not discuss how to explain the reasoning process in CoTs via cognitive neuroscience. Our work provides the first attempt to interpret CoT reasoning from the Hopfieldian perspective.

## 3   PRELIMINARIES

**Large Language Models and Prompting.**   Prompts can take various forms, such as a single sentence or longer paragraphs, and may include additional information or constraints to guide the model's behavior. Let $\mathcal{M} : \mathcal{X} \mapsto \mathcal{Y}$ be an LLM that takes an input sequence $x = (x_1, x_2, \ldots, x_q) \in$

$\mathcal{X}$ and produces an output sequence $y = (y_1, y_2, \ldots, y_m) \in \mathcal{Y}$. The model is typically trained to optimize the conditional probability distribution $pr(y|x)$, which assigns a probability to each possible output sequence $y$ given $x$. To incorporate a prompt $w$ with the input sequence $x$, we can concatenate them into a new sequence $\hat{x} = (w, x_1, x_2, \ldots, x_q)$. The conditional probability distribution $pr(\hat{y}|\hat{x})$ is then computed using $\hat{x}$. Formally, the probability of the output sequence $\hat{y}$ given $\hat{x}$ is:

$$pr(\hat{y}|\hat{x}) = \prod_{i=1}^{m} pr(y_i|y_{<i}, \hat{x}),$$

where $y_{<i}$ represents the prefix of the sequence $y$ up to position $i-1$, and $pr(y_i|y_{<i}, \hat{x})$ denotes the probability of generating $y_i$ given $y_{<i}$ and $\hat{x}$.

**The Hopfieldian View.** In cognitive neuroscience, two prominent perspectives aim to explain cognition: the Sherringtonian view and the Hopfieldian view.[1] The Hopfieldian view focuses on understanding behavior through computation and representation within neural spaces, rather than the specific biological details of neurons, ion flows, or molecular interactions (Hopfield, 1982; 1984; Hopfield & Tank, 1986). It operates at a higher level of abstraction, emphasizing the role of representations and the computations performed on them.

This approach conceptualizes cognition as transformations between representation spaces. At the implementation level, the collective activity of neurons is mapped onto a representation space, which contains a low-dimensional representational manifold. Algorithmically, Hopfieldian computation views these representation spaces as fundamental entities, with movements within or transformations between them as the central operations. The representations themselves are structured as basins of attraction within a state space, and while they are implemented by neural structures (whether individual neurons, neural populations, or other components), the focus is on the dynamics of the system rather than its specific biological mechanisms. Most Hopfieldian models, in practice, center on the activity of neural populations.

A parameter space defines the dimensions of variation within these representational spaces, aligning with quality-space approaches from philosophy, where content is similarly structured. Computations over these representations are understood as dynamic transformations between spaces or shifts within them, characterized by features like attractors, bifurcations, limit cycles, and trajectories. Ultimately, cognitive functions are realized through these dynamic movements within or between representational spaces.

**Linear Representations in Language Models.** Recent investigations into the internal mechanics of LLMs have revealed intriguing properties of their learned representations. Park et al. (2024) posited that high-level semantic features such as gender or honesty could be linearly represented as directions within the model's representation space. This can be illustrated by the well-known word analogy task using a word embedding model (Mikolov et al., 2013). By defining $\mathcal{M}(\cdot)$ as a function of extracting the representations of a given word by a word embedding model, the operation $\mathcal{M}(\text{Spain}) - \mathcal{M}(\text{Madrid}) + \mathcal{M}(\text{Paris})$ often results in an output close to $\mathcal{M}(\text{France})$, where $\mathcal{M}(\text{Spain}) - \mathcal{M}(\text{Madrid})$ can be considered as the representation vector of the abstract "capital of" feature in the embedding space. Concurrently, research on interpretable neurons (Dale et al., 2023; Ortiz-Jiménez et al., 2023; Voita et al., 2024) has identified neurons that consistently activate for specific input features or tasks, suggesting that these features may also be represented as directions in the LLMs' neuron space. For instance, Tigges et al. (2023) use the PCA vector between LLMs' hidden states on instructions "positive" and "negative" to find the sentiment direction in LLMs. Additionally, recent works (Zou et al., 2023; Arditi et al., 2024) show the effectiveness of engineering on language models using these directions. For example, adding multiples of the "honesty" direction to some hidden states has been sufficient to make the model more honest and reduce hallucinations.

## 4 BRIDGING REASONING IN COTS AND THE HOPFIELDIAN VIEW

In this section, we aim to build a bridge between the reasoning process in CoTs and the cognitive brain from the Hopfieldian view. We will particularly associate the main elements (stimuli, neural populations, and representation spaces) in the Hopfieldian view. After understanding these elements, we can leverage the strength of the Hopfieldian view to deepen our understanding of the

---

[1]See Appendix A for an introduction to the Sherringtonian view. For a detailed comparison between these two views, refer to (Barack & Krakauer, 2021) and (Bechtel, 2007).

reasoning process in current CoTs and further improve it. Note that we will leave other elements in the Hopfieldian view, such as attractors and state space, as future work.

**Stimuli and Actions.** Stimuli and actions are key components of how the brain processes information and interacts with the environment. Actions refer to the motor responses or behaviors that result from cognitive processing, which are responses given by LLMs through CoTs.

Stimuli refer to external or internal events, objects, or changes in the environment that are detected by the sensory systems and can influence cognitive processes and behavior. Based on this, we can adopt the term "stimuli" from cognitive science in the context of CoTs to refer to specific prompt text or instructions that trigger CoT reasoning. Specifically, in the zero-shot setting, we define the stimulus as $s_{zero}$ to represent a set of supplementary instructions in the prompt that encourage the model to provide more intermediate reasoning steps before arriving at a final answer. For example, it can be "*let's think step by step*" or "*make sure to give steps before your answer*". In the few-shot setting, the stimulus $s_{few}$ is defined as the sequence of demonstrations $D = \{(\tilde{q}_1, \tilde{a}_1), (\tilde{q}_2, \tilde{a}_2), \dots\}$ in the prompt, where $\tilde{q}_i$ represents the query and $\tilde{a}_i$ is the corresponding response. In the following discussion, we use $s^+$ to indicate that stimuli are included in the model's input and $s^-$ to indicate that no stimuli are added. Note that we avoid using explicitly negative stimuli, such as "*please be careless and answer the following question*", because a well-aligned model would likely refuse to behave in such a manner (Ouyang et al., 2022).

**Neural Populations.** As we mentioned, in the Hopfieldian view, representations are realized by various forms of neural organization, especially populations. Identifying these "neural populations" in CoTs is especially important. In our framework, there are two steps for finding them.

(i) Stimulus Set Designing. Here our goal is to elucidate the sensitivity of LLMs to different CoT prompts with stimuli. Understanding such sensitivity could help us know the neural populations raised from the stimuli. In detail, we construct a prompt set. For each query $q$, we consider two forms of prompts: positive one (with stimuli) as $p^+ = T(s^+, q)$ and negative one (without stimuli) as $p^- = T(s^-, q)$, where $T$ is the prompt template. Specifically, for each query $q_i$, we construct $M$ number of prompts for both of them with different stimuli, which is denoted as $P_i = \{p_1^{i,-}, p_1^{i,+}, p_2^{i,-}, p_2^{i,+}, \dots, p_M^{i,-}, p_M^{i,+}\}$. Such construction is to make our following neural populations less dependent on the specific template form. Thus, in total, we have a stimulus set $P^* = \{P_1, P_2, \cdots, P_N\}$, where $N$ is the number of queries. These contrastive pairs of prompts will be used to identify neural populations given by these stimuli.

(ii) Identifying Neural Populations. Intuitively, the neural populations should be the most influential activation vectors of these prompts or stimuli. In detail, for each prompt in $P^*$, the next step is capturing the network architecture's corresponding neural populations. Since LLMs rely on transformer-based architecture to store distinct representations intended for different purposes, it is crucial to design the extraction process to capture task-specific representations carefully. For a given prompt $p^+$ or $p^-$, we will find the "most representative token", which encapsulates rich and highly generalizable representations of the stimuli. Here we select the last token after tokenizing the prompt, which is based on the observation in Zou et al. (2023) that it is the most informative token for decoder-only or auto-regressive architecture models.

Once the last token position is identified, we can naturally select some of its activations (hidden state) in hidden layers. Previous studies (Fan et al., 2024; Cosentino & Shekkizhar, 2024) have shown that not all layers store important information about reasoning; thus we focus on a subset of them to reduce the computation cost, whose indices are denoted as a set $\mathcal{K}$ (in practice, $\mathcal{K}$ is always the last several layers). Thus, we have a collection of activation vectors. However, since we are focusing on the reasoning of CoT, studying the neural populations raised from the stimuli rather than the whole prompt is more important. Thus, we consider the difference in the activations of pairs of prompts. Specifically, for a pair $(p^+, p^-)$, we can get their activations for all selected layers $\mathcal{K}$: $\{h_k(p^+)\}_{k \in \mathcal{K}}$ and $\{h_k(p^-)\}_{k \in \mathcal{K}}$, where $h_k(p)$ refers to the activation vector of the $k$-th layer for a given input prompt $p$. Then the differences of activations $\{\tilde{h}_k(p)\}_{k \in \mathcal{K}}$ are the neural populations for such stimuli, where $\tilde{h}_k(p) = h_k(p^+) - h_k(p^-)$ represents the most influential information we get from the stimuli for the query. Based on this, for each hidden layer in $\mathcal{K}$, we have the neural population for all queries, which is denoted as

$$h_k^* = \{\tilde{h}_k(P_1), \tilde{h}_k(P_2), \dots, \tilde{h}_k(P_N)\}. \tag{1}$$

**Representation Spaces.** After we have the neural populations for each selected hidden layers, our final goal is to find the representation space. In the Hopfieldian view, the representation of information is thought to occur within low-dimensional space embedded within higher-dimensional neural spaces. Thus, these representation spaces will be the most informative subspaces of the neural populations. Here we adopt the $s$-PCA to find such an $s$ dimensional subspace. Specifically, for the $k$-th layer where $k \in \mathcal{K}$, we perform PCA analysis on $h_k^*$:

$$R_k = \text{PCA}(h_k^*). \tag{2}$$

Then, the space spanned by this eigenvector will be the representation space for this layer. Motivated by the previous linear representation introduced in Section 3, here we set $s = 1$, i.e., we only consider the principal component. Intuitively, this means each representation space will focus on one "concept".

## 5 APPLICATIONS OF HOPFIELDIAN VIEW TO CoTs

In the previous section, we mainly discussed how each element in the Hopfieldian view corresponds to the reasoning in CoTs. From our previous view, we can understand the reasoning process as the movement between these representation spaces. Based on these connections, we can leverage the strength of the Hopfieldian view to improve or further understand CoTs. In this section, we first consider how to localize the reasoning error based on the low dimensional representation spaces. Then, by leveraging the robustness of the Hopfieldian view, we propose a new framework, namely Representation of Thought, that enhances the performance robustness of CoTs.

### 5.1 REASONING ERROR LOCALIZATION

In this task, for a given query, we want to check if there are some reasoning errors in the response by CoTs. If so, we aim to localize these errors. As in the Hopfieldian view, cognition occurs within low-dimensional representation spaces. Reasoning errors can be identified by analyzing the structure of these spaces, such as when certain directions $R_k$ (representing specific cognitive factors) are disproportionately activated or suppressed. This can help localize the source of the error within the cognitive process. Motivated by this, we can leverage the internal structure of spaces we have learned via PCA to locate the reasoning error for a given query in CoTs.

Intuitively, since the reasoning occurs within these representation spaces, if there is a reasoning error in the response, then during the reasoning process, some tokens make the activations (hidden states) of the response far from the corresponding representation spaces. This is because if these activations are far from the spaces, CoTs do not reason the corresponding "concepts" in the response. Motivated by this, our idea is to iteratively check the tokens in the response to see whether they are far from the representation spaces.

Mathematically, for a given prompt $T$ via CoT of query $x$ with its response $y = (y_1, y_2, \cdots, y_m)$, we will iteratively feed the prompt with a part of the response, i.e., $T_i = T \oplus y_{\leq i}$, where $\oplus$ is the string concatenation. If the activations of $T_{i-1}$ are close to while those of $T_i$ are very far from the representation spaces $\{R_k\}_{k \in \mathcal{K}}$ in (2), then we can think the $i$-th token $y_i$ makes an reasoning error. We use the following criterion to access and/or evaluate the quality of the rationale for $T_i$:

$$\text{scores}(T_i) = \text{Mean}(\{\text{scores}_k(T_i)\}_{k \in \mathcal{K}}), \text{ where } \text{scores}_k(T_i) = h_k(T_i)^\top R_k - \delta. \tag{3}$$

Here $\delta$ is the threshold, $\text{scores}_k(T_i)$ is the rationale for the $k$-th representation space, and $\text{scores}(T_i)$ is the average score across all layers in $\mathcal{K}$. When the score is less than 0, it indicates that the activations of prompt $T_i$ are far from the representation spaces. See Algorithm 1 for details.

### 5.2 REPRESENTATION OF THOUGHT

The Hopfieldian view of cognition offers a framework that can potentially be used to control or influence cognitive processes. Specifically, influencing neural populations directly offers a more robust way to control cognition compared to simply providing different stimuli. Firstly, influencing neural populations directly allows the manipulation of the core dynamics of neural state spaces, including attractor states, bifurcations, and transitions between cognitive states. This direct intervention bypasses the variability and unpredictability associated with external stimuli, which depend on the individual's perception, attention, and prior experiences. Moreover, external stimuli are subject to various forms of noise and variability, including sensory processing errors, environmental distractions, and individual differences in interpretation. Direct manipulation of neural populations

---

**Algorithm 1** Reasoning Error Localization

---

**Require:** Prompt $T$ for query $x$; response $y = (y_1, \cdots, y_m)$ of the prompt $T$ via a CoT; threshold $\delta > 0$; representation vectors $\{R_k\}_{k \in \mathcal{K}}$ in (2) with layer set $\mathcal{K}$.

 1: **for** $i = 1, \cdots, m$ **do**
 2:     Denote a new prompt $T_i = T \oplus y_{\leq i}$. Using the same process as in Section 4 to get the activations of $T_i$ in layers in the set $\mathcal{K}$, which are denoted as $h_k(T_i), k \in \mathcal{K}$.
 3:     Calculate $\text{scores}(T_i) = \text{Mean}(\{\text{scores}_k(T_i)\}_{k \in \mathcal{K}})$ in (3).
 4:     **if** $\text{scores}(T_i) < 0$ and $\text{scores}(q_{T-1}) \geq 0$ **then**
 5:         Mark token $y_i$ as a "reasoning error".
 6:     **end if**
 7: **end for**

---

can reduce these sources of noise, providing a cleaner and more consistent pathway to controlling cognitive states.

Our RoT leverages representation spaces' structure to enhance the robustness of reasoning in CoTs. Intuitively, we can manipulate a given query's activations to be closer to the representation spaces to enhance robustness since these spaces are the inherent entities in the reasoning process. After the manipulation, the hidden states will be less dependent on the specific form of the prompt, query, and stimuli but will be more dependent on the intrinsic entities of the reasoning task.

Mathematically, for a given prompt $T$ via CoTs of query $x$. By using a similar procedure as in the Neural Populations section, we can get its neural populations $\{h_k(T)\}_{k \in \mathcal{K}}$. In RoT, motivated by (Zou et al., 2023; Arditi et al., 2024), we can manipulate them by injecting the directions of their corresponding representation spaces to make them closer to these spaces:

$$h'_k(T) = \begin{cases} h_k(T) + \alpha R_k & \text{if } k \in \mathcal{K} \\ h_k(p) & \text{otherwise}, \end{cases} \tag{4}$$

where $h'_k(T)$ denotes the manipulated hidden state, $\alpha$ is a scaling factor controlling the manipulation strength. Its sign should follow the sign of $h_k(T)^\top R_k$.

By directly manipulating neural populations, RoT offers a more precise and interpretable method for influencing the model's output compared to traditional prompt engineering techniques. This approach not only enhances control over the model's behavior but also improves the transparency and predictability of the generation process.

## 6 EXPERIMENTS

In this section, we will perform experimental studies on the above two applications to verify the correctness of our understanding from the Hopfieldian view.

### 6.1 EXPERIMENTAL SETUP

**Datasets.** Our experiments are performed on benchmark datasets for diverse reasoning problems. We consider 6 datasets for 3 different tasks: Arithmetic Reasoning, Commonsense Reasoning, and Symbolic Reasoning. Specifically, for Arithmetic Reasoning, we select GSM8K (Cobbe et al., 2021) and SVAMP (Patel et al., 2021); we study StrategyQA (Geva et al., 2021) and CommonsenseQA (CSQA) (Talmor et al., 2019) for Commonsense Reasoning; lastly, for Symbolic Reasoning, we choose the Coin Flip (Wei et al., 2022) and Random Letter datasets, where the latter one is constructed from the Last Letter dataset (Wei et al., 2022). More details and statistics of the datasets are provided in Appendix B.1.

**LLMs.** We employ Llama-2-7B-Chat (Touvron et al., 2023) and Llama-3-8B-Instruct (Meta, 2024) to evaluate their precision performance (accuracy) both before and after applying RoT to different datasets. Furthermore, we use Llama-2-13B-Chat (Touvron et al., 2023) and Llama-2-70B-Chat (Touvron et al., 2023) to show that our method performs effectively in larger-scale models.

**Baselines.** Since our goal is to analyze the performance and robustness before and after control model reasoning in both zero-shot and few-shot settings, we focus on three baselines in our study: 1) **Base**: as the simplest approach with LLMs for reasoning, feed the model with only one question query. 2) **CoT**$_Z$ (Kojima et al., 2022): the most common zero-shot CoT is employed to provide

Table 1: Results of RoT and CoT based on different LLMs on a variety of reasoning tasks. **Green** indicates an equal or improved accuracy compared to the Base method, while **red** indicates an accuracy decrease. It can be observed that, compared to CoT prompting, RoT achieves more consistent accuracy improvements across a variety of tasks.

| Method | Arithmetic | | Commonsense | | Symbolic | |
|---|---|---|---|---|---|---|
| | GSM8K | SVAMP | CSQA | StrategyQA | Coin Flip | Random Letter |
| Llama-2-7B-Chat | | | | | | |
| Base | 26.00 | 54.00 | 47.75 | 63.62 | 44.80 | 20.33 |
| + $\text{CoT}_Z$ | 26.31 | 46.00 | 43.41 | 62.05 | 52.75 | 24.33 |
| + $\text{RoT}_Z$ | 26.23 | 54.33 | 48.24 | 63.54 | 45.45 | 20.67 |
| Base | 26.00 | 54.00 | 47.75 | 63.62 | 44.80 | 20.33 |
| + $\text{CoT}_F$ | 4.62 | 38.67 | 53.07 | 59.26 | 47.60 | 31.00 |
| + $\text{RoT}_F$ | 25.55 | 56.00 | 48.16 | 63.80 | 45.50 | 20.33 |
| Llama-3-8B-Instruct | | | | | | |
| Base | 73.31 | 80.67 | 72.65 | 65.07 | 68.90 | 44.00 |
| + $\text{CoT}_Z$ | 74.45 | 82.33 | 72.24 | 66.07 | 90.45 | 43.00 |
| + $\text{RoT}_Z$ | 74.83 | 83.33 | 72.89 | 65.24 | 76.35 | 47.67 |
| Base | 73.31 | 80.67 | 72.65 | 65.07 | 68.90 | 44.00 |
| + $\text{CoT}_F$ | 72.02 | 81.00 | 73.63 | 62.75 | 96.50 | 50.67 |
| + $\text{RoT}_F$ | 74.37 | 83.67 | 73.30 | 65.94 | 70.30 | 43.66 |

Table 2: The robust results of our approach and different general baselines with CoT on each task. Bold text indicates optimal results in a single dataset.

| Method | Arithmetic | | Commonsense | | Symbolic | |
|---|---|---|---|---|---|---|
| | GSM8K | SVAMP | CSQA | StrategyQA | Coin Flip | Random Letter |
| Llama-2-7B-Chat | | | | | | |
| $\text{CoT}_Z$ | 5.46 | 11.34 | 6.54 | 6.04 | 8.80 | 14.00 |
| $\text{RoT}_Z$ | **3.02** | **1.32** | **1.64** | **0.70** | **0.30** | **0.68** |
| $\text{CoT}_F$ | 1.44 | **0.00** | 2.78 | **0.48** | 2.70 | 2.00 |
| $\text{RoT}_F$ | **0.08** | 0.67 | **0.00** | 1.88 | **0.00** | **0.00** |
| Llama-3-8B-Instruct | | | | | | |
| $\text{CoT}_Z$ | 33.36 | 85.32 | 2.94 | 5.94 | **13.80** | 18.66 |
| $\text{RoT}_Z$ | **2.58** | **2.66** | **0.82** | **1.14** | 16.40 | **11.34** |
| $\text{CoT}_F$ | **0.23** | **0.33** | 0.74 | 0.26 | 0.45 | 1.00 |
| $\text{RoT}_F$ | 0.37 | 0.34 | **0.33** | **0.26** | **0.45** | **1.00** |

a thought path. 3) **CoT**$_F$ (Wei et al., 2022): directly using some demonstrations before asking a question to LLMs.

**Evaluation Metrics.** We consider the performance of RoT zero-shot ($\text{RoT}_Z$) and few-shot ($\text{RoT}_F$) settings. Besides the utility of performance, which is evaluated by accuracy, we also conducted results on the robustness against forms of prompts. For zero-shot settings, we selected three different specific instructions: (1) *Let's think step by step.* (2) *Let's think about this logically.* (3) *Let's solve this problem by splitting it into steps.* For few-shot settings, we conducted two studies: 1) Using the original order of the given demonstrations, shown in Appendix C.3. 2) Based on experiment 1, we randomly shuffled the order of the demonstrations. Then we use the accuracy difference to consider the robust performance of our approach. Specifically, given a list of accuracy results from $A = \{\tilde{A}_1, \tilde{A}_2, \cdots, \tilde{A}_n\}$ given by different prompts mentioned above, the robust score is calculated by their pairwise difference: $\sum_{i=1}^{n} \sum_{j=i+1}^{n} |\tilde{A}_i - \tilde{A}_j|$. The answer extraction process is based on the methodology outlined by Kojima et al. (2022). Detailed procedures and results are provided in the Appendix C.2.

**Experimental Settings.** If not explicitly stated, in all experiments, we set the number of stimuli prompts $M = 1$, the sample number $N = 128$, and select the samples by high perplexity. At the same time, we set the max new tokens to 512 in the generation stage and pick the last 5 layers to control. We choose $\alpha$ based on the accuracy performance on each dataset. In the reasoning error localization experiment, we set $\delta = 10$. We use float16 to load large language models and employ greedy search as our decoding strategy. All experiments are conducted using one NVIDIA L20 GPU (except Llama-2-70B-Chat which uses three NVIDIA A100 GPUs).

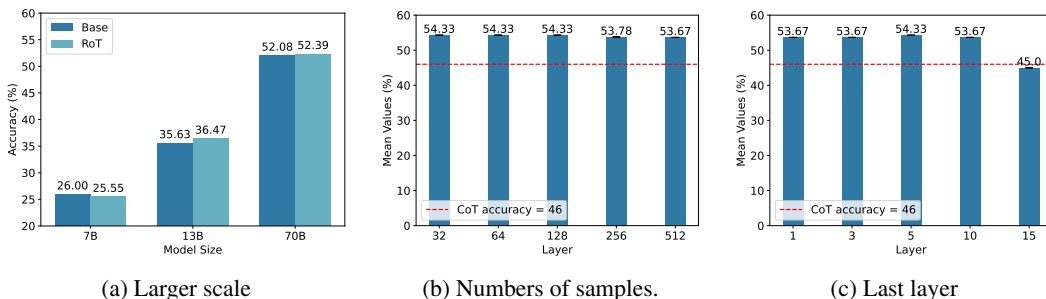

(a) Larger scale          (b) Numbers of samples.          (c) Last layer

Figure 2: Ablation study of our approch. (a) Results on a larger scale on the GSM8K dataset. (b) Results on the number of samples on the SVAMP dataset. (c) Results on the number of selected layers on the SVAMP dataset.

## 6.2 EXPERIMENTAL RESULTS

**Utility Performance.** We first consider the utility performance of RoT. As shown in Table 1, we can see that: 1) The original CoT performs unstable on different tasks. Generally speaking, $\textbf{CoT}_Z$ and $\textbf{CoT}_F$ appear better, but they are lower than **Base** in some datasets, such as the CSQA dataset in the zero-shot scenario, which is consistent with the observation in (Kojima et al., 2022). At the same time, for few-shot, $\textbf{CoT}_F$ performs extremely poorly in the GSM8k dataset because Llama-2-7B-Chat repeats the given demonstrations, resulting in a reduction in the number of valid tokens. Compared to CoTs, our RoT performs strongly in generalization on these datasets but may have lower accuracy in some cases. This is because, in RoT, we add additional directions to the hidden states of the prompt. These manipulations will cause a loss of information regarding the original query, making the accuracy lower. 2) In terms of different models, the Llama-3-8B-Instruct model has been improved more significantly. For example, with Llama-2-7B-Chat as the backbone, $\textbf{RoT}_Z$ is improved by only 0.23 and 0.33 compared with **Base** on the GSM8K and SVAMP datasets, respectively; with Llama-3-8B-Instruct, the improvements are 1.52 and 2.66, respectively. This is primarily because the model is trained on a larger corpus and has learned more knowledge, so the activations contain richer information and can better capture related representations.

**Robustness Analysis.** We also conducted experiments on robustness, and the results are shown in Table 2 (more results are included in Appendix B.2). From this table, we can observe that RoT demonstrates a remarkable advancement over CoT in terms of robustness. We found that CoT methods are very sensitive to prompt design and sometimes fail to output the corresponding response based on the given instruction. However, our RoT extracts more essential information from the representation engineering level, making it more adaptable to various prompts. Note that for Llama-3-8B-Instruct, there are two datasets (SVAMP and Coin Flip) that do not provide robust performance gains. This is because Llama-3-8B-Instruct is a very strong model, while Coin Flip and SVAMP are two relatively easy tasks (as can be seen from the Table 1, the accuracy of CoTs in the SVAMP dataset is greater than 81%, and in the Coin Flip dataset is greater than 90%). These two factors may cause it to over-capture too many irrelevant concepts from the stimuli, thus pointing to the wrong reasoning direction.

**Results on Larger Models.** To further demonstrate the effectiveness of our approach, we conduct research on a larger scale. Specifically, we follow the few-shot settings, and evaluate two larger models (Llama-2-13B-Chat and Llama-2-70B-Chat) on the GSM8K dataset. From Figure 2a, we can see that our approach has improved performance on both 13B and 70B, but the improvement on 70B is smaller. We speculate that this is because the 70B model contains very rich knowledge, and the knowledge contained in a few demonstrations has limited improvement on the model.

**Case Study of Reasoning Error Localization.** We conducted a reasoning error localization experiment. We can calculate the token-level score with Algorithm 1 through our approach. Figure 3 shows that our approach can localize those errors in the response through CoT. In this case, Llama-2-7B-Chat did not really understand the known information in the given question and calculated different units (*hour* and *minute*). Specifically, before calculating the *hour* and *minute* tokens, the scores of the tokens are all greater than zero, indicating no potential errors, while when calculating the *hour* and *minute* tokens, our method detects potential conflicts with previous knowledge and thus obtains a score less than zero. We also show our additional case study in Appendix D.

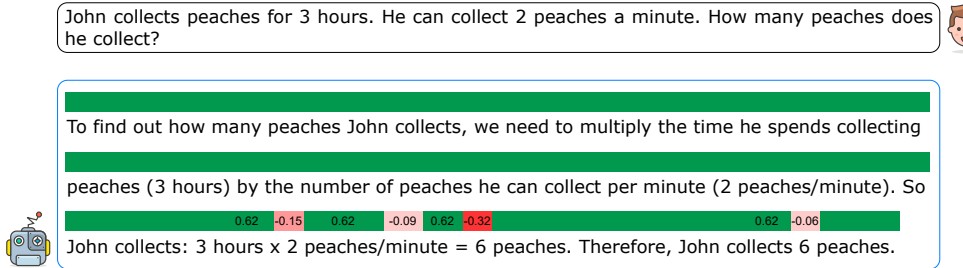

Figure 3: A real case of reasoning error localization by using Llama-2-7B-Chat in a zero-shot scenario on GSM8K using Algorithm 1. The **green** bar indicates that the reasoning snippet is correct, and the **red** bar means that the reasoning snippet may be wrong. The numbers in the bar are the scores calculated by Algorithm 1.

Table 3: Results for different sample selection strategies.

| Model | Perplexity | | Random | | | |
| | Low | High | Seed1 | Seed2 | Seed3 | Seed Avg. |
|---|---|---|---|---|---|---|
| Llama-2-7B-Chat | 23.43 | **25.55** | 25.32 | 25.24 | 25.32 | 25.30 |
| Llama-3-8B-Instruct | 74.22 | **74.37** | 74.52 | 74.37 | 73.92 | 74.27 |

### 6.3 ABLATION STUDY

**Number of Samples.** We conducted an ablation study on how to select samples and how many samples $N$ in the stimulus set for constructing neural populations are sufficient. For the sample selection strategy, we focus on two different strategies and evaluate these on the GSM8K dataset: 1) Random strategy. We randomly select samples in the training dataset using three random seeds. 2) Low Perplexity strategy. We select samples based on low perplexity. 3) High Perplexity strategy. Similar to the low perplexity strategy, we select samples based on high perplexity. As shown in Table 3, we can observe that the high perplexity strategy has better and more generalized performance. This is because high perplexity usually means low confidence in LLMs. Therefore, if a question has a higher perplexity, the question has more latent knowledge information.

For the number of samples $N$, we consider the set $N = \{32, 64, 128, 256, 512\}$ and calculate their average accuracy scores on the SVAMP dataset using three different seeds. From Figure 2b, we can see that the performance is quite stable for different numbers of samples. However, there is still a little decrease when $N$ is large enough. This is because when $N$ is large enough, the representation spaces contain richer information. Thus, adding the directions in (4) will make the query lose more of its query information, causing a lower accuracy.

**Number of Selected Layers.** Here we study the effect of different numbers of selected layers $|\mathcal{K}|$ for neural populations. While LLMs have many layers, such as Llama-2-7B, which contains 32 layers, recent studies have shown that not all layers store important information about reasoning and that this information is usually found in the last layers of the model (Fan et al., 2024; Cosentino & Shekkizhar, 2024). Therefore, we consider the last $L$ layers, where $L = \{1, 3, 5, 10, 15\}$.

In this experiment, we evaluate it with three different seeds. Figure 2c displays the result of average accuracy scores on the SVAMP dataset. From this figure, we can see that the accuracy first increases and then shows a decreasing trend as the number of control layers increases. This is because when the number of layers is very small, each manipulation will correct some of the reasoning errors. However, in RoT we have to manipulate each activation in the layer of the set $\mathcal{K}$, and each manipulation will lose some information about the query. Thus, the accuracy decreases when the number of layers is larger.

### 7 CONCLUSION

In this paper, we proposed a novel framework to explain and understand the fundamental factors behind CoT's success. Specifically, we first connected CoT reasoning and the Hopfieldian view of cognition in cognitive neuroscience. Then, we developed a method for localizing reasoning errors and proposed the RoT framework to enhance the robustness of the reasoning process in CoTs. Experimental results demonstrate that RoT improves the robustness and interpretability of CoT reasoning while offering fine-grained control over the reasoning process.

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

## A  ADDITIONAL PRELIMINARIES

**The Sherringtonian View.** Unlike the Hopfieldian perspective, the Sherringtonian view (Sherrington, 1906; Barlow, 1953) of cognitive explanation emphasizes the importance of direct neuron-to-neuron connections in the brain. This view posits that the primary explanation for cognition lies in the specific interactions between neurons and the computations these neurons perform within well-defined circuits (Mogenson, 2018).

At an algorithmic level, the Sherringtonian view conceptualizes cognition as networks of nodes (neurons) with weighted connections (synapses) between them. In this framework, neurons perform distinct computational transformations on the signals they receive from other neurons in the network. Cognitive processes are described by how individual neurons receive inputs, process these inputs through neural transfer functions, and transmit the resulting signals to connected neurons. Thus, cognition is explained through the computations occurring at the level of individual neurons and the signal flow across their connections.

**Zero-shot CoT.** Zero-shot CoT is a simple but effective chain of thought (CoT) prompting approach proposed by Kojima et al. (2022). It allows language models to generate a step-by-step explanation or thought process to solve problems without requiring prior demonstrations or specific training by simply adding "*Let's think step by step*" before each answer. Specifically, given a query $q$ and a model $\mathcal{M}$ with weights $\theta$, the generation process can be defined as:

$$R = \arg\max pr(\mathcal{R}|q, t; \theta) \tag{5}$$

where $R$ is the response text of the model, which is generated from all potential responses $\mathcal{R}$, and $t$ is a prompt text like "*Let's think sttep by step.*"

**Few-shot CoT.** Similar to zero-shot CoT, few-shot CoT (Wei et al., 2022) is also a prompting technique that gives a few examples with step-by-step reasoning processes to stimulate the model's reasoning ability. Formally, given the query $q$ and a set of demonstrations $D = \{(\tilde{q}_1, \tilde{a}_1), (\tilde{q}_2, \tilde{a}_2), \dots\}$, our aim is to generate a response with intermediate reasoning steps:

$$R = \arg\max pr(\mathcal{R}|D, q; \theta) \tag{6}$$

Table 4: Statistics of the data set.

| Dataset | Task Domain | # Samples | Answer Format |
|---|---|---|---|
| GSM8K | Arithmetic | 1319 | Number |
| SVAMP | Arithmetic | 300 | Number |
| StrategyQA | Commonsense | 2290 | Yes or No |
| CSQA | Commonsense | 1221 | Multiple Choices |
| Coin Flip | Symbolic | 2000 | Yes or No |
| Random Letter | Symbolic | 300 | Letter |

## B  OTHER EXPERIMENTAL DETAILS

### B.1  DATASET

The statistics of the data is shown in Table 4. The details about each data set are as follows:

**Arithmetic Reasoning.** The arithmetic reasoning benchmarks aim to analyze and/or understand the model's mathematical reasoning skills. These include: (i) GSM8K Cobbe et al. (2021), a math word problems benchmark encompassing a variety of reasoning steps; (ii) SVAMP Patel et al. (2021), containing math word problems with multiple structures.

**Commonsense Reasoning.** These data sets aim to analyze the ability of the model on commonsense reasoning tasks. These include: (i) StrategyQA Geva et al. (2021), a commonsense benchmark requiring multi-level strategy to answer the question; (ii) CSQA Talmor et al. (2019) benchmark dataset of multiple-choice questions that require different types of commonsense knowledge to predict the correct answers.

**Symbolic Reasoning.** These data sets aim to test the abilities of the model requiring advanced symbolic capabilities. For this task, we curated two new datasets, as follows. (i) Coin Flip dataset, we employ the data curation strategy of a previous study Wei et al. (2022) using the number of operations as 2, 4 and 7 to come up with the complete dataset; (ii) Random Letter, an advanced version of the last letter concatenation with reference to the previously studied form of word assembly Wei et al. (2022), where 2-4 words are randomly formed and characters are randomly drawn from them, instead of taking the beginning or the end of each word at a fixed point.

Table 5: The detailed robust results in the zero-shot settings.

| Method | Arithmetic | | Commonsense | | Symbolic | |
|---|---|---|---|---|---|---|
| | GSM8K | SVAMP | CSQA | StrategyQA | Coin Flip | Random Letter |
| Llama-2-7B-Chat | | | | | | |
| $CoT_{Z1}$ | 26.31 | 46.00 | 43.41 | 62.05 | 52.75 | 24.33 |
| $CoT_{Z2}$ | 26.23 | 48.33 | 43.90 | 60.52 | 48.35 | 17.67 |
| $CoT_{Z3}$ | 23.58 | 51.67 | 46.68 | 63.54 | 50.10 | 17.33 |
| $RoT_{Z1}$ | 26.23 | 54.33 | 48.24 | 63.54 | 45.45 | 20.67 |
| $RoT_{Z2}$ | 24.72 | 53.67 | 47.91 | 63.58 | 45.50 | 20.67 |
| $RoT_{Z3}$ | 25.09 | 53.67 | 47.42 | 63.23 | 45.35 | 20.33 |
| Llama-3-8B-Instruct | | | | | | |
| $CoT_{Z1}$ | 74.45 | 82.33 | 72.24 | 66.07 | 90.45 | 43.00 |
| $CoT_{Z2}$ | 74.83 | 83.33 | 72.65 | 63.32 | 83.55 | 42.00 |
| $CoT_{Z3}$ | 58.15 | 40.67 | 73.71 | 63.10 | 89.40 | 33.67 |
| $RoT_{Z1}$ | 74.83 | 83.33 | 72.89 | 65.24 | 76.35 | 47.67 |
| $RoT_{Z2}$ | 74.91 | 83.33 | 72.73 | 64.93 | 71.95 | 46.67 |
| $RoT_{Z3}$ | 73.62 | 82.00 | 72.48 | 65.50 | 68.15 | 42.00 |

Table 6: The detailed robust results in the few-shot settings.

| Method | Arithmetic | | Commonsense | | Symbolic | |
|---|---|---|---|---|---|---|
| | GSM8K | SVAMP | CSQA | StrategyQA | Coin Flip | Random Letter |
| Llama-2-7B-Chat | | | | | | |
| $CoT_{F1}$ | 4.62 | 38.67 | 53.07 | 59.26 | 47.60 | 31.00 |
| $CoT_{F2}$ | 3.18 | 38.67 | 50.29 | 59.74 | 50.30 | 29.00 |
| $RoT_{F1}$ | 25.55 | 56.00 | 48.16 | 63.80 | 45.50 | 20.33 |
| $RoT_{F2}$ | 25.63 | 55.33 | 48.16 | 65.68 | 45.50 | 20.33 |
| Llama-3-8B-Instruct | | | | | | |
| $CoT_{F1}$ | 72.02 | 81.00 | 73.63 | 62.75 | 96.50 | 50.67 |
| $CoT_{F2}$ | 72.25 | 80.67 | 72.89 | 67.47 | 95.00 | 50.67 |
| $RoT_{F1}$ | 74.37 | 83.67 | 73.30 | 65.94 | 70.30 | 43.67 |
| $RoT_{F2}$ | 74.00 | 83.33 | 73.63 | 65.68 | 69.85 | 42.67 |

## B.2 DETAILS OF ROBUST EXPERIMENT

We show our detailed results of the robustness in Table 5 and 6. For zero-shot settings, the terms $Z1$, $Z2$, and $Z3$ refer to the use of three different prompts, respectively (as shown in Section 6). For few-shot settings, the terms $F1$ and $F2$ also refer to two different experiments, as shown in Section 6.

## C PROMPTS

### C.1 PROMPT TEMPLATES

Table 7 illustrates the design of stimulus prompts utilized for Chain of Thought (CoT) prompting, distinguishing between zero-shot CoT and few-shot CoT methodologies. In the zero-shot CoT approach, the model is presented with a question devoid of preceding examples, in contrast to the few-shot CoT method, where the model is furnished with multiple exemplars. For each method, the first row is a positive prompt and the second is a negative prompt. Red indicates stimulus token.

Table 7: The stimulus prompting design for CoT-style methods.

| | |
|---|---|
| zero-shot CoT | USER: <question>
ASSISTANT: Let's think step by step. |
| | USER: <question>
ASSISTANT: |
| few-shot CoT | USER: <n different examples>
<question>
ASSISTANT: |
| | USER: <question>
ASSISTANT: |

Table 8: Extraction templates for various tasks.

| Task | Extraction Template |
|---|---|
| StrategyQA | Therefore, the answer (Yes or No) is |
| CSQA | Therefore, among A through E, the answer is |
| Coin Flip | Therefore, the answer (Yes or No) is |
| Random Letter | Therefore, the answer is |

## C.2 ANSWER EXTRACT PROMPTS

The demonstration of our answer extraction method is in Table 8.

## C.3 EXAMPLE PROMPTS FOR FEW-SHOT SETTING

We demonstrate our example prompts for few-shot setting in Table 9, 10, 11, 12 and 13.

## D CASE DEMONSTRATIONS

We show our additional cases on arithmetic, commonsense, and symbolic reasoning tasks in Figure 4, 5, 6, and 7.

Table 9: Few-shot exemplars for the full chain of thought prompt for the arithmetic reasoning task. The highlighted part refers to the reasoning steps.

---

**PROMPT FOR GSM8K AND SVAMP**

**Q:** There are 15 trees in the grove. Grove workers will plant trees in the grove today. After they are done, there will be 21 trees. How many trees did the grove workers plant today?

**A:** There are 15 trees originally. Then there were 21 trees after some more were planted. So there must have been 21 - 15 = 6. The answer is 6.

**Q:** If there are 3 cars in the parking lot and 2 more cars arrive, how many cars are in the parking lot?

**A:** There are originally 3 cars. 2 more cars arrive. 3 + 2 = 5. The answer is 5.

**Q:** Leah had 32 chocolates and her sister had 42. If they ate 35, how many pieces do they have left in total?

**A:** Originally, Leah had 32 chocolates. Her sister had 42. So in total they had 32 + 42 = 74. After eating 35, they had 74 - 35 = 39. The answer is 39.

**Q:** Jason had 20 lollipops. He gave Denny some lollipops. Now Jason has 12 lollipops. How many lollipops did Jason give to Denny?

**A:** Jason started with 20 lollipops. Then he had 12 after giving some to Denny. So he gave Denny 20 - 12 = 8. The answer is 8.

**Q:** Shawn has five toys. For Christmas, he got two toys each from his mom and dad. How many toys does he have now?

**A:** Shawn started with 5 toys. If he got 2 toys each from his mom and dad, then that is 4 more toys. 5 + 4 = 9. The answer is 9.

**Q:** There were nine computers in the server room. Five more computers were installed each day, from Monday to Thursday. How many computers are now in the server room?

**A:** There were originally 9 computers. For each of 4 days, 5 more computers were added. So 5 * 4 = 20 computers were added. 9 + 20 is 29. The answer is 29.

**Q:** Michael had 58 golf balls. On Tuesday, he lost 23 golf balls. On Wednesday, he lost 2 more. How many golf balls did he have at the end of Wednesday?

**A:** Michael started with 58 golf balls. After losing 23 on Tuesday, he had 58 - 23 = 35. After losing 2 more, he had 35 - 2 = 33 golf balls. The answer is 33.

**Q:** Olivia has $23. She bought five bagels for $3 each. How much money does she have left?

**A:** Olivia had 23 dollars. 5 bagels for 3 dollars each will be 5 x 3 = 15 dollars. So she has 23 - 15 dollars left. 23 - 15 is 8. The answer is 8.

---

Table 10: Few-shot exemplars for the full chain of thought prompt for StrategyQA. The highlighted part refers to the reasoning steps.

---

**PROMPT FOR STRATEGYQA**

**Q:** Do hamsters provide food for any animals?

**A:** Hamsters are prey animals. Prey are food for predators. Thus, hamsters provide food for some animals. So the answer is yes.

**Q:** Could Brooke Shields succeed at University of Pennsylvania?

**A:** Brooke Shields went to Princeton University. Princeton University is about as academically rigorous as the University of Pennsylvania. Thus, Brooke Shields could also succeed at the University of Pennsylvania. So the answer is yes.

**Q:** Yes or no: Hydrogen's atomic number squared exceeds number of Spice Girls?

**A:** Hydrogen has an atomic number of 1. 1 squared is 1. There are 5 Spice Girls. Thus, Hydrogen's atomic number squared is less than 5. So the answer is no.

**Q:** Yes or no: Is it common to see frost during some college commencements?

**A:** College commencement ceremonies can happen in December, May, and June. December is in the winter, so there can be frost. Thus, there could be frost at some commencements. So the answer is yes.

**Q:** Yes or no: Could a llama birth twice during War in Vietnam (1945-46)?

**A:** The War in Vietnam was 6 months. The gestation period for a llama is 11 months, which is more than 6 months. Thus, a llama could not give birth twice during the War in Vietnam. So the answer is no.

**Q:** Yes or no: Would a pear sink in water?

**A:** The density of a pear is about $0.6g/cm^3$, which is less than water. Objects less dense than water float. Thus, a pear would float. So the answer is no.

---

Table 11: Few-shot exemplars for the full chain of thought prompt for CSQA. The highlighted part refers to the reasoning steps. There are newlines between the answer choices that are omitted in the table for space reasons.

---

**PROMPT FOR CSQA**

**Q:** What do people use to absorb extra ink from a fountain pen? Answer Choices: (a) shirt pocket (b) calligrapher's hand (c) inkwell (d) desk drawer (e) blotter

**A:** The answer must be an item that can absorb ink. Of the above choices, only blotters are used to absorb ink. So the answer is (e).

**Q:** What home entertainment equipment requires cable?
Answer Choices: (a) radio shack (b) substation (c) television (d) cabinet

**A:** The answer must require cable. Of the above choices, only television requires cable. So the answer is (c).

**Q:** The fox walked from the city into the forest, what was it looking for? Answer Choices: (a) pretty flowers (b) hen house (c) natural habitat (d) storybook

**A:** The answer must be something in the forest. Of the above choices, only natural habitat is in the forest. So the answer is (b).

**Q:** Sammy wanted to go to where the people were. Where might he go? Answer Choices: (a) populated areas (b) race track (c) desert (d) apartment (e) roadblock

**A:** The answer must be a place with a lot of people. Of the above choices, only populated areas have a lot of people. So the answer is (a).

**Q:** Where do you put your grapes just before checking out? Answer Choices: (a) mouth (b) grocery cart (c)super market (d) fruit basket (e) fruit market

**A:** The answer should be the place where grocery items are placed before checking out. Of the above choices, grocery cart makes the most sense for holding grocery items. So the answer is (b).

**Q:** Google Maps and other highway and street GPS services have replaced what? Answer Choices: (a) united states (b) mexico (c) countryside (d) atlas

**A:** The answer must be something that used to do what Google Maps and GPS services do, which is to give directions. Of the above choices, only atlases are used to give directions. So the answer is (d).

**Q:** Before getting a divorce, what did the wife feel who was doing all the work? Answer Choices: (a) harder (b) anguish (c) bitterness (d) tears (e) sadness

**A:** The answer should be the feeling of someone getting divorced who was doing all the work. Of the above choices, the closest feeling is bitterness. So the answer is (c).

---

Table 12: Few-shot exemplars for the full chain of thought prompt for the random letter concatenation task. The highlighted part refers to the reasoning steps.

---

**PROMPT FOR RONDOM LETTER CONCATENATION**

**Q:** Take the first letters of the words in "James Hickman " and concatenate them.

**A:** The first letter of "James" is "J". The first letter of "Hickman" is "H". Concatenating them is "JH". The answer is JH.

**Q:** Take the second letters of the words in "Carl Mccall Bonilla " and concatenate them.

**A:** The second letter of "Carl" is "a". The second letter of "Mccall" is "c". The second letter of "Bonilla" is "o" Concatenating them is "aco". The answer is aco.

**Q:** Take the third letters of the words in "Randy Tanner " and concatenate them.

**A:** The third letter of "Randy" is "n". The third letter of "Tanner" is "n". Concatenating them is "nn". The answer is nn.

**Q:** Take the first letters of the words in "Kenny Kim " and concatenate them.

**A:** The first letter of "Kenny" is "K". The first letter of "Kim" is "K". Concatenating them is "KK". The answer is KK.

---

Table 13: Few-shot exemplars for the full chain of thought prompt for the coin flip task. The highlighted part refers to the reasoning steps.

---

**PROMPT FOR COIN FLIP**

**Q:** A coin is heads up. Ka flips the coin. Sherrie flips the coin. Is the coin still heads up?

**A:** The coin was flipped by Ka and Sherrie. So the coin was flipped 2 times, which is an even number. The coin started heads up, so after an even number of flips, it will still be heads up. So the answer is yes.

**Q:** A coin is heads up. Jamey flips the coin. Teressa flips the coin. Is the coin still heads up?

**A:** The coin was flipped by Jamey and Teressa. So the coin was flipped 2 times, which is an even number. The coin started heads up, so after an even number of flips, it will still be heads up. So the answer is yes.

**Q:** A coin is heads up. Maybelle flips the coin. Shalonda does not flip the coin. Is the coin still heads up?

**A:** The coin was flipped by Maybelle. So the coin was flipped 1 time, which is an odd number. The coin started heads up, so after an odd number of flips, it will be tails up. So the answer is no.

**Q:** A coin is heads up. Millicent does not flip the coin. Conception flips the coin. Is the coin still heads up?

**A:** The coin was flipped by Conception. So the coin was flipped 1 time, which is an odd number. The coin started heads up, so after an odd number of flips, it will be tails up. So the answer is no.

**Q:** A coin is heads up. Sal flips the coin. Raymond does not flip the coin. Is the coin still heads up?

**A:** The coin was flipped by Sal. So the coin was flipped 1 time, which is an odd number. The coin started heads up, so after an odd number of flips, it will be tails up. So the answer is no.

**Q:** A coin is heads up. Conception flips the coin. Kristian does not flip the coin. Is the coin still heads up?

**A:** The coin was flipped by Conception. So the coin was flipped 1 time, which is an odd number. The coin started heads up, so after an odd number of flips, it will be tails up. So the answer is no.

**Q:** A coin is heads up. Inga does not flip the coin. Elanor does not flip the coin. Is the coin still heads up?

**A:** The coin was flipped by no one. So the coin was flipped 0 times. The coin started heads up, and it was not flipped, so it is still heads up. So the answer is yes.

**Q:** A coin is heads up. Ryan flips the coin. Shaunda flips the coin. Is the coin still heads up?

**A:** The coin was flipped by Ryan and Shaunda. So the coin was flipped 2 times, which is an even number. The coin started heads up, so after an even number of flips, it will still be heads up. So the answer is yes.

---

Q: A robe takes 2 bolts of blue fiber and half that much white fiber. How many bolts in total does it take?

To find the total number of bolts needed for the robe, we need to calculate the total amount of blue fiber and white fiber required, and then divide that by the amount of blue fiber required per bolt. The problem states that a robe takes 2 bolts of blue fiber and half that much white fiber.

To find the total amount of blue fiber required, we can multiply the number of bolts by the amount of blue fiber per bolt:
2 bolts x 2 = 4 bolts of blue fiber

To find the total amount of white fiber required, we can multiply the number of bolts by the amount of white fiber per bolt:
2 bolts x 0.5 = 1 bolt of white fiber

Now, we need to divide the total amount of blue fiber by the amount of blue fiber required per bolt to find the total number of bolts needed:
4 bolts of blue fiber / 2 = 2 bolts of blue fiber per bolt

Therefore, it takes 2 bolts of blue fiber and 1 bolt of white fiber to make the robe. ✗

To find the total number of bolts needed for the robe, we need to calculate the total amount of blue fiber and white fiber required, and then divide that by the amount of blue fiber required per bolt. The problem states that a robe takes 2 bolts of blue fiber and half that much white fiber.

So, the total amount of blue fiber required is 2 bolts x 1 bolt = 2 bolts.

The total amount of white fiber required is half of 2 bolts, which is 1 bolt.

+ Control   Therefore, the total number of bolts needed for the robe is 2 + 1 = 3 bolts. ✓

Figure 4: A real case predicted by Llama-2-7B-Chat with zero-shot CoT on the GSM8K dataset. The segment highlighted in blue represents the correct output of the model. The red part shows that the model starts to reason in the wrong direction without control, while the green portion indicates the model reason in the correct direction after adding control.

Q: There are 15 trees in the grove. Grove workers will plant trees in the grove today. After they are done, there will be 21 trees. How many trees did the grove workers plant today?
A: There are 15 trees originally. Then there were 21 trees after some more were planted. So there must have been 21 - 15 = 6.

Q: If there are 3 cars in the parking lot and 2 more cars arrive, how many cars are in the parking lot?
A: There are originally 3 cars. 2 more cars arrive. 3 + 2 = 5.} The answer is 5.
...

Q: Carla is downloading a 200 GB file. Normally she can download 2 GB/minute, but 40% of the way through the download, Windows forces a restart to install updates, which takes 20 minutes. Then Carla has to restart the download from the beginning. How load does it take to download the file?

Let's break this problem down step by step!
Initially, Carla can download 2 GB/minute. To find out how many minutes it would take to download the entire file, we can divide the file size (200 GB) by the download rate (2 GB/minute):
200 GB / 2 GB/minute = 100 minutes

However, Windows forces a restart after 40% of the download is complete. To find out how many GB are left to download, we can calculate 40% of the total file size:
40% of 200 GB = 0.4 x 200 GB = 80 GB

So, Carla has to restart the download from the beginning, and she has 80 GB left to download. To find out how many minutes it will take to download the remaining 80 GB, we can divide the remaining file size by the download rate:
80 GB / 2 GB/minute = 40 minutes.

Adding the initial 100 minutes to the additional 40 minutes, Carla will need a total of:
100 minutes + 40 minutes = 140 minutes to download the entire file.          ✗

Let's break it down step by step:
1. Carla can download 2 GB/minute, so it would normally take her:
200 GB / 2 GB/minute = 100 minutes to download the file.

2. But, 40% of the way through the download, Windows forces a restart. This means she has downloaded:
40% of 200 GB = 80 GB at that point. This means she still has: 200 GB - 80 GB = 120 GB left to download.

3. The restart takes 20 minutes, so Carla loses 20 minutes of download time.

4. After the restart, Carla has to start the download from the beginning, which means she has to re-download the 80 GB she already had. This adds another 80 GB / 2 GB/minute = 40 minutes to the total download time.

Therefore, the total download time is:
100 minutes (original download time) + 20 minutes (restart time) + 40 minutes (re-downloading already downloaded portion) = 160 minutes.          ✓

+ Control

Figure 5: A real case predicted by Llama-3-8B-Instruct with few-shot CoT on the GSM8K dataset. The purple part is an example of input-output pairs given by user. The segment highlighted in blue represents the correct output of the model. The red part shows that the model starts to reason in the wrong direction without control, while the green portion indicates the model reason in the correct direction after adding control.

Q: Do hamsters provide food for any animals?
A: Hamsters are prey animals. Prey are food for predators. Thus, hamsters provide food for some animals. So the answer is yes.

Q: Could Brooke Shields succeed at University of Pennsylvania?
A: Brooke Shields went to Princeton University. Princeton University is about as academically rigorous as the University of Pennsylvania. Thus, Brooke Shields could also succeed at the University of Pennsylvania. So the answer is yes.
...

Q: Is a platypus immune from cholera?

I think I can help you with that!

A: Platypuses are mammals.

Cholera is a bacterial infection that can affect many animals, including mammals. Platypuses do not have a natural immunity to cholera. So, the answer is no. ✗

According to scientific research, platypuses are resistant to cholera.

This is because they have a unique immune system that is adapted to their aquatic environment. Platypuses have a low body temperature, which makes it difficult for many pathogens, including cholera, to replicate and cause disease.

Additionally, their immune system is highly effective at recognizing and eliminating pathogens, and they have a strong response to bacterial infections.

In fact, studies have shown that platypuses are resistant to a range of bacterial infections, including cholera, and that they are able to recover quickly from infections that would be fatal to other animals.

+ Control     So, to answer your question, yes, platypuses are generally immune from cholera! ✓

Figure 6: A real case predicted by Llama-3-8B-Instruct with few-shot CoT on the strategyQA dataset. The purple part is an example of input-output pairs given by user. The segment highlighted in blue represents the correct output of the model. The red part shows that the model starts to reason in the wrong direction without control, while the green portion indicates the model reason in the correct direction after adding control.

Q: A coin is heads up. Ka flips the coin. Sherrie flips the coin. Is the coin still heads up?
A: The coin was flipped by Ka and Sherrie. So the coin was flipped 2 times, which is an even number. The coin started heads up, so after an even number of flips, it will still be heads up. So the answer is yes.

Q: A coin is heads up. Jamey flips the coin. Teressa flips the coin. Is the coin still heads up?
A: The coin was flipped by Jamey and Teressa. So the coin was flipped 2 times, which is an even number. The coin started heads up, so after an even number of flips, it will still be heads up. So the answer is yes.
...

Q: A coin is heads up. verley flips the coin. nytavious does not flip the coin. alsiha does not flip the coin. leroyal does not flip the coin. Is the coin still heads up?

The coin was flipped by Verley. So the coin was flipped 1 time, which is an odd number. The coin started heads up, so after an odd number of flips, it will be tails up.

However, no one else flipped the coin, so the coin was not flipped again. Since the coin was flipped an odd number of times initially, it will still be heads up. So the answer is yes. ✗

**+ Control**

The only person who flipped the coin is Verley. Since the coin was initially heads up, Verley's flip would change the side of the coin.

Therefore, the coin is no longer heads up. ✓

Figure 7: A real case predicted by Llama-3-8B-Instruct with few-shot CoT on the coin flip dataset. The purple part is an example of input-output pairs given by user. The segment highlighted in blue represents the correct output of the model. The red part shows that the model starts to reason in the wrong direction without control, while the green portion indicates the model reason in the correct direction after adding control.

