# OpenReview forum: "Understanding Reasoning in Chain-of-Thought from the Hopfieldian View"
_ICLR.cc/2025/Conference — ICLR 2025 Conference Withdrawn Submission_

### Official Review · Reviewer_Qtgm · 2024-10-16

**Soundness:** 2
**Presentation:** 2
**Contribution:** 2
**Rating:** 3
**Confidence:** 3

**Summary:**

This paper aims to understand and enhance chain-of-thought reasoning from the perspective of cognitive neuroscience.
The author references the Hopfieldian View in cognitive science to establish a connection between chain-of-thought reasoning and cognitive science.
Subsequently, the author proposes RoT (Representation-of-Thought) by modeling the processes in chain-of-thought reasoning from a Hopfieldian perspective.

As described by the author, the Hopfieldian perspective explains the process of how the brain responds to external stimuli and produces behaviors from both computational and representational angles.
Specifically, the paper maps the elements of the Hopfieldian view to the elements of chain-of-thought reasoning.
The article models several key elements of the Hopfieldian view and CoT, and the correspondences between them are as follows:
- Action in Hopfieldian: Actions refer to the motor responses or behaviors that result from cognitive processing.
- Action in CoT: The CoT reasoning responses of LLMs
- Stimuli in Hopfieldian: Stimuli refer to external or internal events, objects, or changes in the environment that are detected by the sensory systems and can influence cognitive processes and behavior.
- Stimuli in CoT: The prompts. In zeroshot CoT, it's corresponding to instruction. And in fewshot CoT, it is the demonstration.
- Neuron populations in Hopfieldian: Hopfieldian models the cognition from neuron perspevtive, especially populations of neurons.
- Neuron Populations in CoT: Here they are the neural presentations in LLMs, specially, the presentation of the last token after tokenization.
- Representation Space in Hopfieldian: In the Hopfieldian view, the representation of information is thought to occur within low-dimensional space embedded within higher-dimensional neural spaces.
- Representation Space in CoT: This paper uses s-PCA to reduce the dimensiona1 of the aforementioned neuron populations.

Then, the author employs the Hopfieldian perspective to enhance chain-of-thought reasoning.
They iteratively uses the representation space in CoT reasoning to identify errors in reasoning.
Experiments were conducted on arithmetic reasoning, commonsense reasoning, and symbolic reasoning tasks.

**Strengths:**

1. This paper aims to understand the process of chain-of-thought reasoning from the perspective of brain cognitive neuroscience, using the Hopfieldian view as the theoretical foundation of cognitive science, and mapping the basic concepts of chain-of-thought reasoning to those of cognitive science.
2. They attempt to enhance chain-of-thought reasoning from a cognitive science perspective. The authors use neural representation space to explore errors in chain-of-thought reasoning, thereby enhancing reasoning.

**Weaknesses:**

1. This article extensively introduces concepts from cognitive neuroscience, but when it comes to establishing a connection between cognitive neuroscience and chain-of-thought reasoning, it feels rather forced. The article merely makes an imprecise connection between concepts from cognitive neuroscience and the behavior of modeling chain-of-thought reasoning.
2. The method proposed in the paper has less connection to concepts in cognitive neuroscience and is more of an heuristic algorithm. For this reason, the paper seems to overclain its correspondences to cognition science.
3. The writing of the article, in an attempt to incorporate cognitive neuroscience, packages a relatively simple and easily understandable algorithm using the Hopfieldian perspective, thereby increasing the difficulty of understanding.
4. The experiments do not demonstrate the advantages of the proposed method, RoT. In many cases, RoT is inferior to vanilla CoT, and in some cases, it is even weaker than non-chain-of-thought reasoning.

**Questions:**

Please see weaknesses

---

### Official Review · Reviewer_kwC6 · 2024-11-02

**Soundness:** 1
**Presentation:** 1
**Contribution:** 1
**Rating:** 3
**Confidence:** 4

**Summary:**

This work provides an analysis of Chain-of-Thought (CoT) drawing on concepts from cognitive neuroscience, and uses this analysis to develop a novel perturbation method to improve the effectiveness of CoT. Experiments are performed on reasoning benchmarks, and several analyses are performed (ablation study, qualitative case study, etc) to better understand the method.

**Strengths:**

- The topic is certainly important, and I encourage the authors to continue working on this. It would be very beneficial to better understand how prompting methods such as CoT can improve reasoning capabilities.
- It is great to see work that takes inspiration from cognitive neuroscience, as this is a field with lots of potential insights for better understanding contemporary AI models.
- A number of experiments are performed to better understand the method, including an ablation study.

**Weaknesses:**

- The primary weakness is that the proposed method does not seem to produce reliable improvements. There are no statistical tests to assess whether any of the results are statistically meaningful, and the differences in performance between the proposed method and the baselines is very small. In many cases the baselines actually perform better. Overall, the results give the impression that none of the methods investigated (including the proposed method) make much of a difference on these tasks.
- Though I applaud the effort to take inspiration from cognitive neuroscience, the particular insights that are explored in this work are rather generic, and also somewhat incorrectly attributed to the field as a whole. The inspiration for this work seems to be a particular perspective piece (Barack & Krakauer, 2021) that explores two broad theoretical frameworks in neuroscience, one of which emphasizes the properties of individual neurons and low-level mechanisms (dubbed the 'Sherringtonian' view), while the other emphasizes the higher-level mathematical properties of populations of neurons (dubbed the 'Hopfieldian' view). This is an interesting perspective, but there are a number of issues with the way in which the ideas are adopted and applied to the present work:
   - First, these terms are presented as if they are standard terminology in cog neuro, but as far as I can tell they were actually introduced in the perspective piece. These are not terms that cognitive neuroscientists use to refer to these views.
   - The 'hopfieldian' view, as discussed in the present work, basically amounts to the claim that the population level is the right level of analysis for understanding neural phenomena. This is reasonable, but it's probably the default view within the field of deep learning research. This is a very generic insight, and there is no clear connection between this general concept and the specific methods proposed in this work.
- The case study methodology is not systematic. Only a few examples are presented, and there is no description of how these examples were selected. It is unclear if these results are representative.

**Questions:**

How were the case study examples selected? Is it possible to perform a systematic analysis over a larger set of examples?

---

### Official Review · Reviewer_2M4N · 2024-11-03

**Soundness:** 2
**Presentation:** 3
**Contribution:** 2
**Rating:** 5
**Confidence:** 2

**Summary:**

This paper attempts to draw a connection between chain-of-thought reasoning in artificial intelligence and Hopfieldian cognitive science, introducing a concept referred to as RoT (Reasoning of Thought). While the authors present their arguments systematically, there are significant concerns regarding the validity of their claims.

**Strengths:**

1. The paper is well-structured and provides a comprehensive overview of the topics discussed.
2. The general idea of incorporating cognitive science perspectives into LLM reasoning is thought-provoking and could spark further research.

**Weaknesses:**

Well, the main concern about the paper is the idea itself. From my perspective, the idea demonstrates far-fetched connections. The authors propose a link between CoT reasoning and Hopfieldian cognitive science, yet the evidence provided is tenuous.

For instance:
(1) The authors suggest that the dynamics of Hopfield networks can directly inform CoT processes in LLMs without adequately addressing the fundamental differences between cognitive processes and machine learning paradigms, especially the decoding-only transformer architecture and the generation process of CoT prompts.
(2) Key terms and concepts from cognitive science are employed without a rigorous explanation of how they translate to LLM CoT reasoning. For instance, the analogy made between cognitive retrieval processes and LLM operations lacks empirical backing and may lead to misinterpretations of both fields.
(3) Additionally, the paper does not sufficiently engage with existing literature that critiques the application of cognitive science principles to AI, which could have strengthened their argument.

Given the speculative nature of the claims and the insufficient evidence for the proposed connections, I think the paper is marginally below the acceptance threshold. A more rigorous exploration of these ideas, with stronger empirical support, would be necessary for meaningful contributions to the field.

Also, I would consider raising my score if the author provide convincible theoretical evidence in the rebuttal process.

**Questions:**

See the weaknesses

---

### Official Review · Reviewer_ueQB · 2024-11-04

**Soundness:** 2
**Presentation:** 2
**Contribution:** 2
**Rating:** 3
**Confidence:** 4

**Summary:**

It is interesting to explore the validity of CoT. This paper combines the cognitive mechanism of CoT and Hopfield, and thus proposes a more robust RoT based on the representation space of the neural population, and proves the better interpretability of RoT relative to CoT by localizing the inference error.

**Strengths:**

1. for the first time, the effect of neural populations, i.e., activation vectors, generated by different prompt stimuli, on inference is studied in representation space and its relevance is demonstrated
2. ablation experiments demonstrate that for problems with higher perplexity, the introduction of RoT produces a more significant improvement;
3. a new error localization method is proposed and validated

**Weaknesses:**

1. The gradient color of the deviation in representation space in the CoT part in Fig. 1 is easily confused with the gradient color in the neural network schematic, because blue is used to show the strength of the motion strength in the left figure, while the same color in the neural network frame of right figure is used to distinguish the model layers.
2. Table 1 COT_F is only 4.62 on GSM8K dataset. it may be a mistake, it should be 24.62.
3. only 4 LLAMA models are evaluated in this paper. I would recommend testing models in different structures to validate your conclusion

**Questions:**

1. it remains unclear whether RoT is more effective for models with fewer parameters or if its benefits extend to larger models with substantial intrinsic knowledge. In the experiment part, the author argues the 70-billion-parameter model shows less accuracy improvement with RoT compared to the 13-billion-parameter model, attributing this to the limited additional knowledge that a small demonstration can provide relative to the model’s intrinsic knowledge. However, the claim that the llama-8b-instruct model performs well due to learning more knowledge than the 7-billion-parameter model contradicts the earlier assertion. Does that mean RoT is only effective within a certain model size range or a certain level of intrinsic knowledge? Without knowing this, it would be hard to validate the reliability of RoT and encourage other researchers to further generalize RoT in other LLMs.

I would recommend
1) conducting additional experiments to determine if there is an optimal model size range for RoT effectiveness.
2) discussing the implications of their findings for applying RoT to different sizes of language models.
3) testing RoT on a wider range of model sizes or comparing its effectiveness across different types of tasks.


2.  This article categorizes null stimuli as negative input, which might be overly simplified. The distinction between “negative” and “null” stimuli deserves a clearer rationale. This article argues explicitly that negative prompts offer less help but I don't think that it is convincing enough to simplify the 'negative stimuli' into 'null stimuli'. From my understanding, many possible cues can be viewed as negative stimuli like distracting content, ambiguous synonyms, etc. For example,

p = " There are 15 trees in the grove. Grove workers will plant trees in the grove today. After they are done, there will be 21 trees. How many trees did the grove workers plant today?"
p+ = p + "Let's Think Step by Step"
p- =  " There are 15 trees in the grove. Grove workers will plant trees in the grove today. After they are done, there will be 21 trees. Those workers are not willing to work today. (Distracting irrelevant description) How many trees did the grove workers plant today?"

or

p- =  " There are 15 plants (synonynum) in the grove. Grove workers will plant trees in the grove today. After they are done, there will be 21 trees. Those workers are not willing to work today. (Distracting irrelevant description) How many trees did the grove workers plant today?"

---

### Note · Authors · 2024-11-24

I have read and agree with the venue's withdrawal policy on behalf of myself and my co-authors.